# Recent Advances in Aflatoxins Detection Based on Nanomaterials

**DOI:** 10.3390/nano10091626

**Published:** 2020-08-19

**Authors:** Chunlei Yan, Qi Wang, Qingli Yang, Wei Wu

**Affiliations:** 1College of Food Science and Engineering, Qingdao Agricultural University, Qingdao 266109, China; chunleiyan233@163.com (C.Y.); 17854231141@163.com (Q.W.); 2State Key Laboratory of Bio-Fibers and Eco-Textiles, Institute of Biochemical Engineering, College of Materials Science and Engineering, Qingdao University, Qingdao 266071, China

**Keywords:** aflatoxins, biosensors, nanomaterials, detection

## Abstract

Aflatoxins are the secondary metabolites of *Aspergillus flavus* and *Aspergillus parasiticus* and are highly toxic and carcinogenic, teratogenic and mutagenic. Ingestion of crops and food contaminated by aflatoxins causes extremely serious harm to human and animal health. Therefore, there is an urgent need for a selective, sensitive and simple method for the determination of aflatoxins. Due to their high performance and multipurpose characteristics, nanomaterials have been developed and applied to the monitoring of various targets, overcoming the limitations of traditional methods, which include process complexity, time-consuming and laborious methodologies and the need for expensive instruments. At the same time, nanomaterials provide general promise for the detection of aflatoxins with high sensitivity, selectivity and simplicity. This review provides an overview of recent developments in nanomaterials employed for the detection of aflatoxins. The basic aspects of aflatoxin toxicity and the significance of aflatoxin detection are also reviewed. In addition, the development of different biosensors and nanomaterials for aflatoxin detection is introduced. The current capabilities and limitations and future challenges in aflatoxin detection and analysis are also addressed.

## 1. Introduction

Mycotoxins are secondary metabolites of fungi that widely exist in nature. Mycotoxins ingested, inhaled or absorbed through the skin can cause functional decline, disease and even death in humans and animals. Among more than 400 different mycotoxins, aflatoxins are the most toxic. Aflatoxins are a kind of secondary metabolite produced by fungi such as *Aspergillus flavus* and *Aspergillus parasiticus* through the polyketone pathway, which is the general name of a group of compounds with similar structure that have strong toxicity, carcinogenicity, mutagenicity and teratogenicity. At present, there are more than 20 kinds of aflatoxins and derivatives [1,2,3]. They are composed of three elements, C, H, and O, and their chemical structures are very similar. They all contain a difuran ring and a coumarin in the molecule. Among various aflatoxins, the four most important ones are aflatoxin B1, B2, G1 and G2, also called AFB1, AFB2, AFGl and AFG2, among which AFB1 is the most toxic. AFB1 will be hydroxylated by liver microsomal enzymes in cattle after intake of AFB1-contaminated feed and converted into AFM1, which is then excreted in the milk of cattle. The results showed that the toxicity of aflatoxin B1 was 10 times of that of potassium cyanide and 68 times of that of arsenic [4]. Its carcinogenicity is 70 times that of dimethylnitrosamine. The International Agency for Research on Cancer (IARC) classified AFB1, AFB2, AFG1 and AFG2 as category 1 carcinogens and AFM1 as a category 2B carcinogen in 1993 [5,6]. Aflatoxins are mainly ingested by humans and animals through dietary channels [7]. In 2020, it was reported that AFB1 exposure is a major risk factor for human primary liver cancer; it can also induce gastric cancer, renal cancer, rectal cancer, breast cancer and other tumors [8]. The AFB1 toxin shows strong accumulation in the liver after entering animals, causing liver hemorrhage, fatty degeneration, bile duct hyperplasia and so on, even leading to liver cancer. Trace amounts of AFB1 can cause acute poisoning death, carcinogenesis and other extremely serious harms in humans or animals.

Aflatoxins widely exist in nature and can contaminate a variety of crops and foods, such as peanuts, cereals, corn, rice, nuts, feed, milk and so on, and may be introduced during food harvest, processing, storage and transportation. People are exposed directly through the ingestion of food contaminated with aflatoxins and indirectly through the consumption of animal products. Long-term intake of food contaminated by aflatoxins, even if the concentration of aflatoxins is very low, threatens human health, causing liver damage and even death because of accumulation in the body. Because aflatoxins are very heat stable, they are very difficult to destroy once formed; in addition, according to the existing agricultural and food production processing conditions, aflatoxins in food and feed cannot be completely avoided, and thus it is particularly important to develop and establish sound monitoring and detection procedures for aflatoxins. For this reason, most countries have formulated the limit standards of aflatoxins in food. Among them, the European Union has the most stringent standards [9], which set the maximum limits of AFB1 in cereals, peanuts and dried fruits for direct human consumer goods at 2 μg/kg, the maximum limits of AFB1, AFB2, AFG1 and AFG2 combined at 4 μg/kg, and the maximum limit of AFM1 in milk at 0.05 μg/kg.

Accurate, rapid and convenient detection of aflatoxins in food is a necessary prerequisite for scientific and effective treatment of contaminated food. The Association of Official Analytical Chemists (AOAC) and the European Committee for Standardization (CEN) have published 47 aflatoxin detection methods. At present, the detection methods for aflatoxins are divided into chemical analysis, biological identification, instrumental analysis and immunoassays. Thin layer chromatography (TLC), one of the chemical analysis methods, was used as the standard method for the detection of aflatoxins by the AOAC in 1990 [10]. However, it is a semiquantitative method with low sensitivity, requires direct contact with standard sample and poses a high risk of exposure to the assayer. Bioassays are qualitative analyses, usually used only as corroboration [11,12]. Instrumental analysis is a national standard detection method with accurate and reliable results and good repeatability, but the instrument is expensive to purchase and maintain; requiring complex pretreatment and long analysis time, it is not suitable for rapid detection of a large number of samples in real time, and it is difficult to promote at the grass-roots level [13,14]. Generally, high performance liquid chromatography (HPLC) is still the most common and widely used method for the detection of aflatoxins in the laboratory at present [15,16,17], but detection methods based on immunology have been increasingly used in the laboratory to detect aflatoxins because of their speed, use of simple equipment, high specificity, simple operation and low cost. At present, the immunological methods used for aflatoxin detection are enzyme-linked immunosorbent assay (ELISA) [18,19], lateral flow immunoassay (LFIA) [18,20], fluorescence immunoassay [21,22], chemiluminescence immunoassay [23,24], and immunosensor assay [25,26]. The common aflatoxin detection methods are summarized in Table 1.

In recent years, with the rapid development of nanotechnology, a variety of nanomaterials with special functions have been designed and prepared. Nanomaterials are those materials that have at least one dimension on the nanometer scale (0.1–100 nm) in three-dimensional space. The discovery of nanomaterials marks a great step forward in the field of exploring the microcosmic world and promotes the progress of chemical analysis, and nanomaterials enjoy a reputation as “the most promising materials in the 21st century”. Due to their special structure, nanomaterials have good surface effect, high reactivity and small size effect and show many useful characteristics in terms of catalytic, electrochemical, thermal and optical properties. Therefore, nanoparticles will have broad application prospects in the chemical industry, life sciences, environmental protection and other fields [27,28,29,30,31,32,33,34,35,36]. To further improve the sensitivity and simplicity of aflatoxin analysis and detection, more types of nanomaterials are being used, such as metal nanomaterials, carbon nano-materials, metal oxide and hydroxide nanomaterials. The properties of these nanomaterials and their applications in the detection of aflatoxins are introduced and discussed (Table 2). When the scale of materials is reduced to the nanometer level, nanomaterials can show many unique properties, and their combination with biosensors will be expected to further improve the performance of biosensors. Therefore, the development of nanomaterials as a new type of biosensor has been favored by many researchers. The use of nanomaterials can help to improve the sensitivity of biosensors and shorten the sample preparation and experimental detection time; therefore, nanomaterials are deeply studied and widely used in the quantitative detection of aflatoxins. In this paper, the research progress on nanomaterials for the detection of aflatoxins in recent years is reviewed. In addition, the research progress on the combination of nanomaterials and biosensors for the detection of aflatoxins is also introduced. The limitations and future challenges of nanomaterial detection technology are also discussed.

## 2. Metal Nanomaterials for Aflatoxins Detection

Metal nanomaterials can be combined with a variety of biological macromolecules, have strong adsorption characteristics, do not affect normal biological activities, and can be used to detect a variety of enzyme activities. At the same time, metal nanomaterials also have good biocompatibility and are easy to combine with biomolecules in food safety detection. The increase in available binding sites for biomolecules on the surface is also a major advantage, so that metal nanoparticles can bind to more biomolecules, such as antibody, aptamer, enzyme and so on. In addition, the unique fluorescence and electrical properties of metal nanomaterials are the main basis for food analysis. According to the different emission or absorption wavelengths of the corresponding metal nanomaterials, aflatoxins can be quantitatively and qualitatively analyzed. When the particle size of these materials is small enough, they will have a quantum size effect and show optical, electrical, chemical and catalytic activities [67,68,69].

Zhang et al. [70] prepared a semiquantitative immunochromatographic strip that can be assayed visually. A monoclonal antibody against aflatoxin B1 labeled with gold nanoparticles was sprayed on the binding pad, establishing two areas on the nitrocellulose membrane, namely, the detection area and the control area. The control area was fixed with rabbit anti-mouse secondary antibody. Unlike the traditional test strip, in this strip, the detection area contained three detection lines fixed to recognize a specific amount of AFB1–bovine serum albumin (BSA). The detection limit of the strip was 0.06 ng/mL, and the cut-off thresholds of the three detection lines were 0.125, 0.5 and 2.0 ng/mL, respectively. The improved strip can provide a better reference for toxin content and realize semiquantitative detection of analytes. Song et al. [71] established a LFIA method for simultaneous quantitative or semiquantitative determination of AFB1, ZEA (zearalenone), DON (vomitoxin) and their analogues in grain samples. The detection limits of AFB1, ZEA, and DON were 0.03, 1.6 and 10 μg/kg, respectively. Anfossi et al. [72] prepared a strip for quantitative detection of the four major aflatoxins. The detection limit of AFB1 in maize was 1 μg/kg.

Xu et al. [43] designed a label-free optical biosensor for the determination of AFB1. GNRs were used as a sensing platform, and the GNR–AFB1–BSA conjugate aggregated with free antibody after mixing, which led to a significant change in the absorption intensity. At the same time, the presence of AFB1 molecules in the samples leads to the dispersion of nanorods, which is the result of competitive immune reaction with antibodies. Under the condition of high ionic strength, there is no need to add any stabilizer as the material has high stability. UV–Vis absorption intensity is used as a sensing index, and dynamic light scattering (DLS) measurement is used as another sensing tool. The biosensor system can detect AFB1 in the range of 0.5–20 ng/mL and an LOD of 0.16 ng/mL was achieved. The system has a good sensitivity for the determination of absorbance. The recoveries of AFB1 in actual peanut samples ranged from 94.2% to 117.3%. Therefore, the nanobiosensor provides a high sensitivity, good selectivity and simple operation method for the rapid screening of toxins in agricultural products and food. ARATI et al. [37] constructed an electrochemical immunoassay method for AFB1 by using gold nanoparticle (AuNP)-modified PEDOT-GO. First, the electrocatalytic behavior of the modified glassy carbon electrode (GCE) was studied by cyclic voltammetry (CV) and electrochemical impedance spectroscopy (EIS); then, it was found that the current signal was linear for AFB1 concentration in phosphoric acid buffer in the range of 0.5–20 ng/mL and 20–60 ng/mL, and the sensitivity of the assay for these ranges was 0.989 and 0.397 μA/(ng/mL), respectively. The limit of detection (LOD) was 0.109 ng/mL. The method was also confirmed to be applicable to corn AFB1 assay, with an LOD of 0.09 ng/mL. In this method, GO was not only used as a fixed platform but also facilitated the rapid transmission of electrons (Figure 1).

The electrochemical biosensor is a sensing platform that evaluates changes in electrical parameters (current, impedance, conductivity, voltage, or potential signal) through biochemical reactions between a biometric element and a target analyte [73]. Simple equipment, high sensitivity and easy miniaturization have always been the key research object of analytical researchers [74,75]. In recent years, electrochemical biosensors combined with nanomaterials have been vigorously developed because they can overcome the shortcomings of poor selectivity and low sensitivity in the original sensor detection methods. A novel electrochemical aptamer sensor based on AuNPs and AFM1 aptamer (Apt) and aptamer CS was constructed for AFM1 detection by Seyed et al. [42]. In the absence of the target molecule (AFM1), the Apt hairpin structure is not opened, and thus it cannot be combined with CS-modified AuNPs. In the presence of AFM1, the structure of the Apt hairpin is opened, and the Apt/AFM1 complex is formed. Therefore, the 5′ end of Apt is exposed and able to hybridize with CS on the surface of AuNPs. A small amount of methylene blue was added to increase the sensitivity of the sensors. A high concentration of methylene blue can be enriched on the electrode surface and generate strong electrochemical signals. However, when AFM1 is not present, Apt will retain its hairpin structure. As a result, CS-modified AuNPs could not combine with Apt, resulting in a low concentration of methylene blue on the electrode surface that could only produce weak electrochemical signals (Figure 2). The detection limit of this system was 0.9 ng/L, and it has been successfully applied to the detection of real samples (milk and serum).

rGO is a kind of graphene nanomaterial that can partially reduce the oxygen-containing functional groups of GO lamellae. Its structure is similar to those of GO and graphene, but compared with GO, it has faster electron transport velocity, and thus it is more widely used in the chemical analysis of aflatoxins. SHU et al. [76] complexed rGO functionalized with cobalt tetraphenyl porphyrin (CoTPP) and platinum nanoparticles (PtNPs) with AFB1 antibody to form an anti-AFB1/PtNPs/CoTPP/rGO complex. A novel competitive immunoassay-based electrochemical method for the determination of AFB1 was developed based on DPV using the reduction current of H_2_O_2_ catalyzed by PtNPs/CoTPP/rGO nanocomposites. The LOD was 1.5 pg/mL. GO, conducting polymer (2,5-di-(2-thienyl)-1-pyrrole-1-(p-benzoic acid), DPB) and gold nanoparticles were modified on the surface of gold electrode [40], and then the aflatoxin antibody was conjugated to the conductive polymer film by 1-(3-dimethylaminopropyl)-3-ethylcarbodiimide hydrochloride (EDC)/N-hydroxysuccinimide (NHS). Finally, chitosan-containing ionic liquid was modified on the electrode surface to prepare the electrochemical impedance immunoassay for AFB1 in food samples. Among them, graphene and gold nanoparticles ensure the rapid transmission of electrons, ionic liquid provides a mild microenvironment for antibodies, and electrochemical impedance testing ensures a high sensitivity of detection. The LOD was 1.0 × 10^−15^ mol/L. The recoveries for peanut, rice, milk, flour and soybean were in the range of 96.3−101.2%. The method has good precision and accuracy. ALTHAGAFI et al. [41] constructed an ultrasensitive AFB1 detection method based on the change in current value caused by antigen–antibody reaction by depositing gold nanodot-modified rGO nanomaterials on ITO conductive glass and immobilizing antibody on the composite electrode. The LOD was 6.9 pg/mL.

## 3. Metal Oxides and Hydroxides Nanomaterials for Aflatoxins Detection

Due to the lack of adjacent atoms around the surface atoms (surface effect), the surface of metal compounds such as nanomaterials is rich in suspended bonds. This unsaturated property gives excellent chemical activity and has certain catalytic properties for specific substrates, which can provide the advantage of signal amplification. Therefore, they can improve the analytical performance of biosensors with low LOD and short deposition time. Compared with metal nanomaterials, metal oxides and hydroxides have better stability and lower cost, so they are also a kind of nanomaterial that is widely used in aflatoxin detection [77,78]. Recently, due to the superior physicochemical properties (such as high mechanical strength, large surface area, high catalytic performance and abundant electronic properties), metal oxides and hydroxides nanomaterials have been regarded as promising candidates for identifying various target analytes through various EC technologies. Srivastava et al. [38] constructed an Au@rGO/ITO immunoelectrode by depositing rGO on ITO glass modified with AuNPs. The detection range was 0.1–12 ng/mL, and an LOD of 0.1 ng/mL was achieved, which was lower than the detection limit for rGO alone. Recently, the same group modified rGO with Ni nanoparticles (Ni NPs) by hydrazine hydrate and deposited the composite on ITO glass to construct a rGO-Ni NPs/ITO composite electrode. Differential pulse voltammetry (DPV) was used for AFB1 detection. The sensitivity of the method was 129.6 μA/[(ng·mL^−1^)·cm^2^] at 1–8 ng/mL. The high sensitivity was mainly attributed to the synergistic effect between rGO and Ni NPs [39]. Srivastava et al. [44] used an electrophoretic deposition technique to deposit rGO on ITO conductive glass. AFB1 antibody was covalently crosslinked to rGO/ITO by EDC/NHS, and its electrochemical behavior was studied by CV and EIS (Figure 3). When the concentration of AFB1 was in the range of 0.125–1.5 ng/mL, the peak current was linear with the concentration of AFB1, the correlation coefficient was 0.99, and an LOD of 0.15 ng/mL was achieved; the sensitivity was 68 μA/[(ng·mL^−1^)·cm^2^].

Singh et al. [45] designed an electrochemical sensor by immobilizing Sm_2_O_3_ nanorods and nickel nanoparticles on the surface of the electrode. The anti-AFB1 antibody was used as the recognition element and can detect 10–700 pg/mL AFB1 in food. As a common metal oxide, Fe_3_O_4_ has been widely investigated for its low cost, nontoxicity, ease of production and storage, and so on. Chauhan et al. [46] developed a reusable electrochemical quartz crystal microbalance (QCM) immunosensor for the detection of AFB1 in food. Monoclonal anti-AFB1 antibody was immobilized on a gold-coated quartz crystal electrode as a capture antibody, while the rabbit IgG detection antibody was bound to Au-Fe_3_O_4_ nanoparticles. The bioelectrode can be regenerated. In the experiment, AFB1 in the corn sample was captured on the electrode surface by the capture antibody while the free antigen was washed away, and the detection antibody continued to react with AFB1 in the sample, forming a sandwich complex on the electrode surface. The amount of AFB1 in the sample was calculated by detecting the magnitude of the signal generated by the label on the complex, and the detection range was 0.05–5 ng/mL.

## 4. Carbon Nanomaterials for Aflatoxins Detection

Carbon nanomaterials mainly include carbon quantum dots (QDs), fullerenes, carbon nanotubes, carbon nanofibers, graphene and their derivatives. Carbon nanomaterials not only have nano effects but also have excellent electrical properties (such as wider potential window and fast electron transmission speed), strong adsorption properties, good chemical stability, good shape controllability and biocompatibility, ease of functionalization and other characteristics that lay the foundation for their application in analysis and detection [79]. The application of carbon nanomaterials in electrochemical analysis is often not limited to a single role and function but most of the time involves the effective superposition of multiple functions to complete the improvement of aflatoxin analysis performance. GAN et al. [47] constructed an ultrasensitive electrochemiluminescence (ECL) device based on magnetic graphene oxide extraction of AFM1 and antibody-labeled CdTe QDs-carbon nanotube (CNT) composite to detect AFM1 in milk. This method not only utilizes the good adsorption characteristics of carbon nanomaterials but also suggests that CNTs can amplify the ECL signal. GO-Fe_3_O_4_ magnetic composite material was used as an adsorbent to extract AFM1 from milk. The AFM1 antibody was combined with the CdTe QDs–CNTs complex to form a signal marker, which was then immobilized on a screen-printed carbon electrode (SPCE). The content of AFM1 was determined by the intensity of the electrochemiluminescence signal of an immune complex formed by antigen–antibody specific recognition reaction. The LOD for the AFM1 was at a concentration of 0.3 pg/mL, which was much higher than that of ELISA.

### 4.1. Graphene Nanomaterials

Graphene is a two-dimensional nanomaterial with atomic thickness and a hexagonal honeycomb structure composed of SP^2^ hybrid carbon atoms and is the basic component unit of all carbon materials. Graphene has fast electron transfer rate (200,000 cm^2^/(V·s)), excellent mechanical properties, high thermal conductivity, large specific surface area and good biocompatibility. GO is a derivative of graphene that is produced by different oxidation of graphene sheets. The preparation of GO is usually from graphite oxide, which is more convenient than graphene. GO is easy to couple with electrochemically active substances and biomolecules because of the different oxidation degree and the different number of oxygen-containing functional groups. Due to the oxygen-containing functional groups, it has good biocompatibility and hydrophilicity, which ensures that the structure and activity of a variety of biomolecules are not destroyed after coupling and is convenient for the construction of aflatoxin analysis methods with different detection modes.

Srivastava et al. [54] prepared GO/Au electrode by homogeneously dispersing graphene oxide solution on the surface of gold (Au) electrode. The AFB1 antibody was successfully immobilized on the surface of the electrode by using EDC as coupling agent and NHS as activator, then immobilized with BSA. The BSA/anti-AFB1/GO/Au immunocomplex electrode was constructed by blocking the nonspecific sites. The charge transfer resistance (Rct) of the electrode as a function of AFB1 concentration in phosphate buffer solution containing potassium ferricyanide was studied by EIS. The results showed that there was a wide linear range between Rct and AFB1 concentration of 0.5–5 ng/mL, and the LOD was 0.23 ng/mL. Rijian Mo et al. [80] proposed a novel biosensor based on GO-modified polyacrylic acid (PAA) film for the detection of AFB1 through π–π stacking with the AFB1 aptamer. The aptamer of AFB1 was covalently immobilized on the channel surface of PAA membrane, and then graphene oxide was added to bind the aptamer. Upon introducing the negatively charged graphene oxide and aptamer, the negative charge of PAA nanochannels increases, causing steric hindrance. In the presence of AFB1, aptamer-specific recognition and binding of AFB1 occurred, graphene oxide was separated from the surface of the PAA membrane, and the charge density and steric hindrance were reduced. As a result, the amount of Fe(CN)_6_^3−^ added through the nanochannel increases, and the current response increases. The sensor has good selectivity for AFB1, with an LOD of 0.13 ng/mL and a wide linear range from 1 ng/mL to 20 ng/mL. Goud et al. [81] constructed an electrochemical aptamer sensor for the detection of AFB1 using GO and aptamer labeled by MB redox probe as a signal amplification platform. The functionalized graphene oxide was immobilized on the SPCE, and then the MB-labeled aptamer was immobilized on the SPCE through a carbodiimide amide bond using hexamethylenediamine (HMDA) as a spacer. When AFB1 interacts with the target molecule, the double-stranded form of the aptamer will be converted to the G-quadruplex form, which is close to the electrode surface and carries out rapid electron transfer. Increasing the concentration of the target analyte also increases the amount of quadruplex structure formed, and subsequently, more and more aptamers labeled with MB are closer to the electrode surface, resulting in an increased electrochemical signal. In this design, the graphene oxide layer increases the conductivity and catalytic characteristics of the sensor system, which helps to improve the sensitivity of electrochemical signals to target analytes. The linear response range of AFB1 was 0.05–6.0 ng/mL, and the LOD was 0.05 ng/mL.

Recently, GQDs have also been used in the detection and analysis of aflatoxins based on the above detection mode. SHADJOU et al. [59] invented an AFM1 electrochemical analysis method in which GQDs, α-cyclodextrin and silver nanoparticles (AgNPs) were successively deposited on GCE to prepare a ternary composite film, and the AFM1 detection method was established by linear sweep voltammetry (LSV). In this method, α-cyclodextrin as a conducting medium, GQDs as a stabilizer, and AgNPs as an electrocatalyst jointly contributed to the target detection and signal amplification. Under the optimal condition, the detection range was 0.015–25 mmol/L, and the limit of quantitation was 2 μmol/L. This method can be used for the detection of AFM1 in milk without any treatment. Wang et al. [56] prepared PPy/PPa/rGO nanocomposite film on GCE by constant current polymerization by using the composite solution of reduced graphene oxide, PPy and PPa and fixed the antibody on the surface of a PPy/PPa/rGO/GCE electrode by covalent coupling for the ultrasensitive detection of AFB1. A sensitive AFB1 detection method was established by exploring the relationship between the change in Rct of an antibody-modified electrode before and after the test of the substance to be tested and the concentration of AFB1 in the substance to be tested. This method has a good response to AFB1 in the detection range of 10 fg/mL–10 pg/mL. At the same concentration level, the respective crossreaction rates of AFB2, AFG1 and AFG2 were 5.0%, 30.6% and 20.1%. The crossreaction rates of vomitoxin and ochratoxin were lower than 1.0%.

With the development of the application of graphene nanomaterials in aflatoxin analysis and detection, researchers have also tried to combine precious metals, ionic liquids or conductive polymers with graphene nanomaterials to improve the performance of aflatoxin electrochemical analysis. Shi [58] introduced nanogold-modified PANI/graphene nanocomposites and fixed them on a gold electrode to form an Au/PA-NI/G/Au electrode detection sensor. The electrochemical analysis method for AFB1 was established by immune reaction and square wave cyclic voltammetry (SWV). The detection range was 0.05–25 ng/mL, and the LOD was 0.034 ng/mL. GELETA et al. [57] constructed an electrochemical method for aflatoxin analysis based on rGO/molybdenum disulfide/polyaniline nanocomposites mixed with chitosan and coated on the surface of glassy carbon electrode, after which the electrode was modified with gold nanoparticles and AFB1 aptamer in turn. The blocking of electron exchange caused by the change of spatial structure between toxin and aptamer leads to the change in current response. The detection of AFB1 was established by the DPV method. A wide linear range from 1.0 × 10^−17^ g/mL to 1.0 × l0^−15^ g/mL and a detection limit of 2 × 10^−18^ g/mL showed the good specificity of this method. KHOSHFETRAT et al. [55] reported an electrochemiluminescence method for the detection of AFM1 in milk based on aptamer technology and GO. First, the thiolated aptamer of AFM1 was fixed on gold-coated magnetic nanoparticles (GMNPs) to form the Apt–GMNPs complex. Luminol-functionalized silver nanoparticle-decorated graphene oxide (GO–L–AgNPs) successfully combined with Apt–GMNPs through a π–π interaction between GO and unpaired bases on aptamers to form an Apt–GMNPs–GO–L–AgNPs complex. When AFM1 is present, the aptamer reacts with the toxin, leading to the separation of the GO–L–AgNPs, which leads to a change in the electrochemiluminescence signal. Based on the above principle, the authors established an analytical method with the analytical range of 5–150 ng/mL and LOD of 0.01 ng/mL; this method was used in milk with high reproducibility.

### 4.2. Carbon Nanotube Nanomaterials

Carbon nanotubes (CNTs) are a kind of carbon nanomaterial that consists of coiled graphite sheets with a layered structure. Carbon nanotubes can be divided into SWCNTs and multiwalled carbon nanotubes (MWCNTs) according to the number of wall lamellae. Carbon nanotubes (CNTs) have some special electrical properties, such as metallicity, semiconductivity, high electrical conductivity and good electrocatalytic activity. As a result, CNTs become an excellent electron transport material that can greatly accelerate the transfer of electrons between electroactive materials. Singh et al. [49] prepared c-MWCNTs/ITO composite electrodes by one-step electrophoretic deposition of c-MWCNTs on ITO glass. A BSA/anti-AFB1/MWCNTs/ITO immune electrode was prepared by covalently coupling aflatoxin monoclonal antibody to the composite electrode via EDC and NHS and blocking its nonspecific active sites with BSA. The association constant of the electrode was 0.0915 ng/mL, indicating that the electrode has a strong affinity for AFB1. The results showed the method had high sensitivity in the linear range of 0.25–1.375 ng/mL (LOD = 0.08 ng/mL).

Zhang et al. [50] designed an indirect, competitive electrochemical immunoassay for AFB1 based on SWCNTs/chitosan (SWCNTs/CS). First, the SWCNTs/CS/GCE electrode was prepared by dropping SWCNTs/CS nanocomposite on a glassy carbon electrode surface. AFB1–BSA was immobilized on the electrode surface by EDC/NHS reaction, and the redundant sites were blocked by BSA. Then, AFB1 antibody was immobilized on the electrode by competing with the AFB1–BSA immobilized on the electrode surface and the free AFB1 in the test solution. After that, the alkaline phosphatase-labeled secondary antibody (AP-anti-antibody) was immobilized onto the electrode surface by reacting with the primary antibody. Finally, the composite electrode was immersed in a solution containing α naphthyl phosphate (α NP). Alkaline phosphatase catalyzed the hydrolysis of α-NP to produce electrochemical signals to achieve the purpose of indirect detection of AFB1 concentration. Differential pulse voltammetry results showed that the current density decreased linearly with the logarithm of AFB1 concentration in the range of 0.01–100 ng/mL, with an LOD of 3.5 pg/mL. The detection limits for AFB1 in actual cornmeal samples were as low as 13.5 pg/mL, which is much lower than the limit standard set by most countries.

Yu et al. [51] modified the GCE electrode surface with a complex composed of MWCNTs, [BMIM]PF_6_ ionic liquid and aflatoxin antibody. Upon specific immunoreaction with aflatoxins, the aflatoxin concentration can be detected through the linear relationship between electron transfer resistance and AFB1 concentration before and after immunoreaction. The range of detection was 0.1–10 ng/mL, and the LOD was 0.03 ng/mL. This method has been applied to the determination of aflatoxins in olive oil. Zhang et al. [48] used PDDA as the dispersant and charge regulator of carbon nanotubes. Carbon nanotubes/PDDA/palladium-gold nanoparticles (CNTs/PDDA/Pd-Au) were prepared. AFB1 was immobilized on a gold electrode by immobilizing AFB1 antibody and BSA on the electrode in turn. AFB1 was quantitatively analyzed by immunoreaction and differential pulse voltammetry. The detection range of this method was 0.05–25 ng/mL, and the LOD was 0.03 ng/mL.

Noble metals can be compatible with a variety of biological molecules and have good electrocatalytic activity. Therefore, electrochemical analysis of aflatoxins through methods involving carbon nanotubes has been reported frequently. Li et al. [52] coupled an AFO sol–gel/MWCNT/Pt electrode by embedding anatoxin-oxidase (AFO) in a silica sol–gel solution with a multiwalled carbon nanotube modified platinum electrode to catalytic oxidation of AFB1. The linear range of the method was 3.2 × 10^−9^–721 × 10^−9^ mol/L, the sensitivity was 0.33 × 10^2^ A/[(mol·L^−1^)·cm^2^], and the LOD was 1.6 × 10^−9^ mol/L. Wang et al. [53] developed a molecularly imprinted electrochemical method using a stepwise approach for the detection of AFB1. Au/Pt bimetallic nanoparticles (AU/PtNPs) were electrodeposited on glassy carbon electrode modified with MWNTs, and then AFB1 was removed with hydrochloric acid to obtain a MI–POPD–Au/PtNPs–MCNTs–GCE electrode. CV, DPV and EIS were used to investigate the electrochemical performance of the samples. The detection range was l × 10^−10^–l × 10^−5^ mol/L, and the LOD was 0.03 nmol/L. The method was proven to be applicable to the determination of AFB1 in gutter oil.

### 4.3. Other Carbon Nanomaterials

Other carbon nanomaterials, such as carbon nanohorns, mesoporous carbon, carbon dots and carbon nanospheres, have also been used for aflatoxin analysis because of their different microstructures and unique physical and chemical properties. For example, mesoporous carbon materials have been widely used in the field of catalysis, especially in electrode materials and electrocatalysts, due to their high conductivity, physical and chemical stability and high specific surface area. Carbon nanospheres have regular geometry, high specific surface area and excellent biocompatibility and are often used to prepare new multienzyme labels for detection signal amplification to improve the sensitivity of analysis.

MONDAL et al. [82] synthesized poly(methyl methacrylate) (PMMA) and polyacrylonitrile (PAN) on the surface of silicon wafers by electrospinning and spin coating techniques. Then, platinum nanoparticles were modified on the micropore channels by thermal decomposition, and aflatoxin monoclonal antibody was covalently crosslinked to the surface of PtNPs/micropore carbon electrode by EDC/NHS. Based on the change in impedance caused by immunoreaction, the method for the analysis of aflatoxins was established. The detection range was 1.0 × 10^−12^–0.1 × 10^−6^ g/mL, and the LOD was 1.0 × 10^−12^ g/mL. Microporous carbon electrodes not only provide a microenvironment for the deposition of PtNPs and antigen–antibody reaction but also improve the electron transport rate, thus promoting the improvement of electrochemical performance. XU et al. [60] constructed an electrochemiluminescence immunoassay for AFB1 based on carbon nanohorns (CNHs) and magnetic nanomaterials. CNHs and luminol-functionalized Fe_3_O_4_ nanofibers (L-F_3_O_4_-NFs) were modified on the surface of magnetic glassy carbon electrode (MGCE). The AFB1 antibody was covalently coupled to the L-F_3_O_4_-NFs/CNHs/MGCE composite electrode. The nonspecific sites were blocked by BSA. The ECL signal produced by luminol and immunoreaction was used to detect AFB1. CNHs with excellent conductivity, biocompatibility, large specific surface area and variable porosity can greatly enhance the electrochemiluminescence signal of luminol. F_3_O_4_ nanomaterials can adsorb a large amount of the signaling material luminol, which is also a substrate for the immobilization of antibodies, and thus a large number of antibodies can be enriched on the surface of MGCE. The LOD was 0.02 ng/mL, and the detection range was 0.05–200 ng/mL.

## 5. Other Nanomaterials for Aflatoxins Detection

Magnetic nanomaterials are a new type of superparamagnetic materials developed with the appearance of nano materials in 1980s. Magnetic nanomaterials can be quickly separated from the solution under the action of external magnetic field, which greatly simplifies the separation operations such as filtration and centrifugation, and overcomes the problems of high cost, difficult recovery and secondary pollution of traditional adsorbents. Wang et al. [61] constructed a competitive colorimetric immunoassay for the detection of AFB1 using biologically functionalized MBs and gold nanoparticles (GNPs). MBS modified by AFB1–BSA was used as capture probe, which was specifically bound to anti-AFB1 antibody labeled by GNP through the immune response. In the presence of AFB1, it has a competitive inhibitory effect on this specific binding. The supernatant containing unbound GNPs was directly detected by UV–Vis after magnetic separation. After optimization, this method can detect AFB1 linearly in the range of 20–800 ng/L. The LOD was 12 ng/L. The recoveries of corn samples ranged from 92.8% to 122.0%. The immunoassay provides a simple, rapid, specific and economical method for toxin detection in the field of food safety. The team also established a simple, sensitive and economical method to detect AFB1 (Figure 4). AFB1–BSA combined with modified magnetic beads was used as capture probe, and gold colloid coated with anti-AFB1 antibody was used as detection probe to perform immune recognition and signal transduction of AFB1. Quantitative measurement was carried out by ultraviolet visible spectrum. Under the optimized conditions, the linear range was 0.01–1 ng/mL, and the LOD was 7 pg/mL. This method provides a good prospect for the sensitive detection of other mycotoxins and organic pollutants [62].

Zhang et al. [63] reported a fluorescent aptamer sensor for simultaneous detection of OTA and AFB1. The aptamer of OTA (Ap1) and AFB1 (Ap2) immobilized on the surface of MBs hybridized with signal probe 1 (Sp1) and signal probe 2 (Sp2), respectively, to form the Aps–Sps duplex structure. Because the stability of the toxin target–Aps is higher than that of the Aps–Sps duplex structure, in the presence of OTA and AFB1, G–tetrad and AFB1–Aps complexes are, respectively, formed by binding with the corresponding Aps, and Sp1 and Sp2 are released. After magnetic separation, the released Sps synthesizes DNA scaffold-silver nanoparticles with different photoluminescence bands. The fluorescence intensity was significantly enhanced by Zn(II). The fluorescence intensity of the sensor was linear with the concentration of OTA and AFB1 (0.001–0.050 ng/mL). The LOD of OTA and AFB1 was 0.2 pg/mL and 0.3 pg/mL, respectively. The recoveries of OTA in wheat, rice and corn were 88.50% ± 6.90~113.50% ± 5.30. The recoveries of AFB1 were 88.10% ± 5.70 ~ 116.80% ± 4.90. The development of this method opens up a new means of simultaneously detecting more types of mycotoxins. ELISA is one of the most widely recognized and widely used spectroscopic analysis methods, but it usually requires several hours of analysis time. To address this problem, a series of magnetic ELISAs for aflatoxin analysis have been developed [83,84,85,86]. Madalina Tudorache et al. [83] proposed a magnetic particle-based ELISA (mp-ELISA) for the immunoassay of AFB1. Anti-aflatoxin B1 antibody was immobilized on the surface of the magnetic particles. The magnetic particles and anti-AFB1 antibody can be easily manipulated by placing a permanent magnet close to the microplate wall. As a solid carrier, magnetic particles accelerate the interaction between immunoreagents and promote the rapid separation of immune complexes, thus reducing the analysis time. The effects of immobilization method and antibody type on the sensitivity of mp-ELISA were examined. Aflatoxin B1 was determined under the optimized parameters.

Wang et al. [65] bound AuNPs/DNA composite to aptamer-modified MNPs by DNA hybridization. After AFB1 was replaced, the AuNPs/DNA nanocomposites were treated with terminal deoxynucleotidyl transferase (TDT) to extend the DNA. The PAPDI was combined with this complex through electrostatic interaction. The dissociation of PAPDI causes a change in the fluorescence intensity, which is used to quantify AFB1. This design is relatively simple and modifies PAPDI without a tag, which increases the fluorescence signal. At the same time, the method uses dozens of probe molecules to generate an amplified fluorescence signal. In addition, the use of DNA amplification by DNA strand extension further enhances the fluorescence intensity and improves the sensitivity of the method. The detection limit was 0.01 nM (3.1 pg/mL). It is expected to be among the most reliable AFB1 fluorescence detection platforms. Yao et al. [64] developed an ultrasensitive chemiluminescence sensor for the selective detection of AFB1 with the oxidation of luminol using horseradish peroxidase (HRP) as the catalyst and HCR as the signal enhancer. First, the capture probe (CP) on the surface of the magnetic beads hybridizes with the aptamer of the target to form a double-stranded DNA structure. When AFB1 is present, the aptamer binds to AFB1, and the CP probe is exposed to the bead. Subsequently, the sticky end of the CP triggers the HCR, forming a long tandem containing alternating hairpins 1 (H1) and 2 (H2). H1 is repeat labeled with a biotin moiety at its end and is linked to streptavidin-labeled HRP (SAHRP). Then, a large amount of HRP catalyzes the luminol–H_2_O_2_ redox reaction to produce strong chemiluminescence (CL) emissions. Under the optimum experimental conditions, there was a good linear correlation between the concentration of CL and AFB1 in the range of 0.5–40 ng/mL. The LOD was 0.2 ng/mL. The method has been applied for the determination of AFB1 in peanut and milk samples.

Fluorescence signal detection methods have the advantages of high sensitivity, low detection limits and short determination times, and thus biosensors based on fluorescence signals have attracted wide attention. Fluorescence biosensors mainly measure the change in fluorescence signal caused by the interaction between recognition elements and targets [87,88,89,90,91]. Recognition elements are easily adsorbed by nanomaterials and are increasingly widely studied and applied in the field of aflatoxin detection. A fluorescent aptamer sensor for AFB1 detection has been developed by Sabet et al. [66]. In this system, QDs and AuNPs act as donor and acceptor, respectively. QDs–aptamer is adsorbed on the surface of AuNPs through the electrostatic interaction between aptamer and AuNPs, which results in fluorescence quenching. After the introduction of AFB1, the QDs–aptamer binds to AFB1 and forms the QDs–aptamer target complex, which is far from the surface of the AuNPs, and the fluorescence of the QDs can be restored. Under the optimal conditions, the LOD of the sensor was 3.4 nmol/L. The method has been successfully applied to the analysis of AFB1 in rice (recovery: 103~108%) and peanut (recovery: 104.5~108.0%) samples with satisfactory results. Colorimetric signal detection is an analysis method involving a color change that can be seen with the naked eye, which has the advantage of simple, fast and semiquantitative visual detection. Seok et al. [92] developed a colorimetric sensor for AFB1-induced structural changes in a DNA enzyme based on the DNA enzyme-heme/aptamer complex. In this system, two DNA-cleaving enzymes (α and β) hybridize to the complementary region of the AFB1 aptamer to form a G-tetrad and oxidize 2,2-azino-bis(3-ethylbenzothiazoline-6-sulfonicacid) (ABTS) in the presence of heme under the catalysis of H_2_O_2_. In the presence of AFB1, aptamer-specific recognition of AFB1 leads to the dissociation of the DNA enzyme-heme/aptamer complex, resulting in visible color change. The linear range of the sensor was 0.1–1.0 × 10^4^ ng/mL, and the LOD was 0.1 ng/mL. The recoveries of AFB1 in corn samples were in the range of 93.96–104.95%, and the sensor was stable (RSD was less than 6%).

## 6. Conclusions

Food safety and human health are closely linked, and food quality and safety is also a pressing issue that many people have paid attention to and attached importance to in recent years. At present, there is still much room for the development of aflatoxin detection methods. The detection flux needs to be improved. However, most methods can only detect a single component, which obviously limits their practical application. It is of great significance to develop methods that can detect multiple aflatoxins or biotoxins at the same time for ensuring food safety. At present, most of the electrochemical sensing devices used for aflatoxin detection are still in the laboratory research stage, and it is difficult to detect samples in real time as conveniently and quickly as is possible with commercial kits or blood glucose meters. Practical, portable, reusable and rapid detection methods and analytical devices need to be further studied. Existing methods use many detection modes, such as identification technology, immune methods, aptamer technology, molecular imprinting and so on. As a new food analysis method, most of the detection schemes of nanomaterials detection technology are in the laboratory stage, which shows that the whole operation process is not stable enough and is still immature. The main reason is that the preparation conditions of nanomaterials are still strict, and the detection process requires the use of large-scale instruments. Therefore, the application of nanomaterials for detection is still limited. With the continuous improvement of the preparation scheme of nanomaterials and the simplification of the detection technology, lightweight and simple instruments can be used for detection while leaving the large-scale instruments in the laboratory; this is the aspect that needs to be continuously improved. At the same time, to detect many components simultaneously, it is necessary to find suitable experimental methods and reagents for specific substances. In real life, however, detection may need to be performed in a large number of substances that are unpredictable at present. This urgently requires efficient and versatile detection methods that can not only preliminarily determine the presence certain substances in food, such as metals, active proteases, harmful pathogens, carbohydrates, and fats, but also determine the general range of their concentration. Nanomaterials can accurately detect the specific components and contents of these substances, which is critical in food detection. Therefore, improving the detection methods of nanomaterial technology, expanding the detection range and simplifying the detection equipment are the directions of future research.

## Figures and Tables

**Figure 1 nanomaterials-10-01626-f001:**
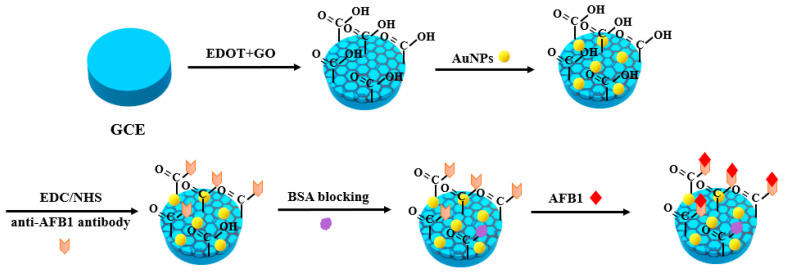
Scheme of BSA/anti-AFB1/AuNPs/PEDOT-GO/GCE-based immunosensor for AFB1 detection.

**Figure 2 nanomaterials-10-01626-f002:**
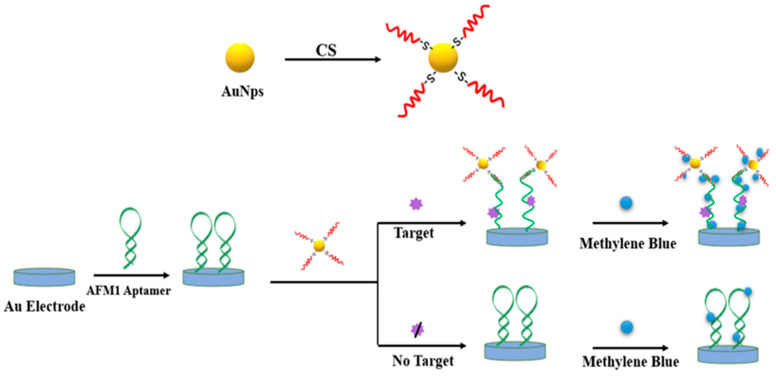
An electrochemical aptamer sensor for AFM1 detection.

**Figure 3 nanomaterials-10-01626-f003:**
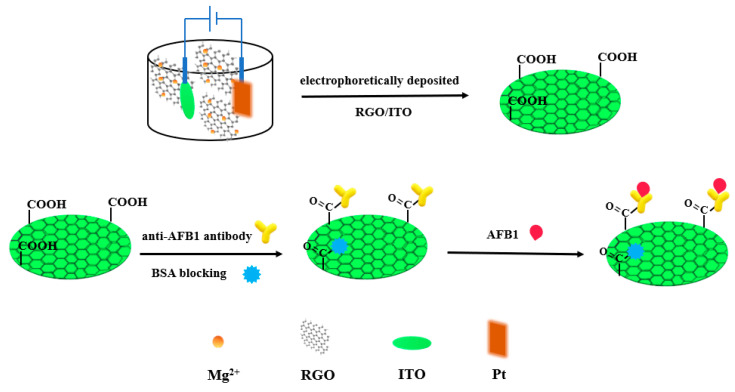
Scheme of electrophoretically deposited rGO platform for AFB1 detection.

**Figure 4 nanomaterials-10-01626-f004:**
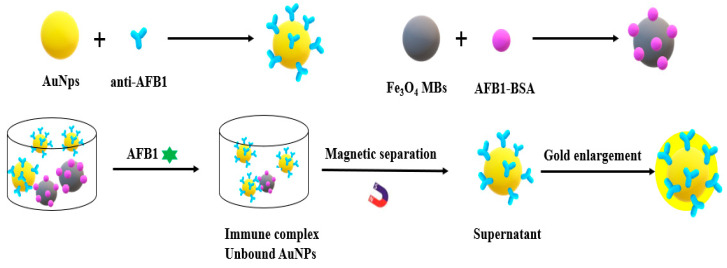
Schematic illustration of the homogeneous gold staining for amplified optical detection of AFB1.

**Table 1 nanomaterials-10-01626-t001:** The common detection methods of aflatoxins.

Detection Method	Advantages	Disadvantages
TLC	Simple equipment, low cost and easy operation.	Cumbersome steps, poor sensitivity, high detection limit and reagent are harmful to operators.
HPLC	Good repeatability, low detection limit and high sensitivity.	Needs derivation, complex operation and high instrument cost.
UPLC	Fast detection speed, short experimental period, no derivative and high sensitivity.	High instrument cost.
LC-MS	Simple pretreatment, high selectivity and multi-component analysis.	Complex equipment operation and high instrument cost.
ELISA	Large number of samples can be analysed simultaneously, high sensitivity and accuracy, does not require extensive sample cleanup.	Short reagent life, higher false positive probability.
LFIA	Fast detection speed, low cost, easy operation, simple equipment, short experimental period.	Poor repeatability, difficult to quantify, poor sensitivity.

Note: TLC: Thin layer chromatography; HPLC: high performance liquid chromatography; UPLC: Ultra performance liquid chromatography; LC-MS: Liquid chromatography-mass spectrometry; ELISA: enzyme-linked immunosorbent assay; LFIA: lateral flow immunoassay.

**Table 2 nanomaterials-10-01626-t002:** Different nanomaterials used for aflatoxins detection.

Type of Nanomaterials/NPs		Properties	Receptor Molecules	Target	Detection Signal	Linear Range	LOD	Real Sample	Ref.
Metal nanomaterials	AuNPs	High extinction coefficients, Good stability and conductivity, Good photoelectric performance, Good biocompatibility, High specific surface area, Easy to modify.	Antibody/PEDOT	AFB1	Electrochemical signal	0.5–20 ng/mL 20–60 ng/mL	0.09 ng/mL	Corn	[37]
	AuNPs		Antibody	AFB1	Electrochemical signal	0.1–12 ng/mL	0.1 ng/mL	-	[38]
	Ni NPs		Antibody	AFB1	Electrochemical signal	1–8 ng/mL	0.16 ng/mL	-	[39]
	AuNPs		Antibody/DPB	AFB1	Electrochemical signal	-	1.0 × l0^−15^ mol/L	Peanut, rice, milk, flour, soybean	[40]
	AuNPs		Antibody	AFB1	Electrochemical signal	100 ng/mL–1 pg/mL	6.9 pg/mL	Peanut	[41]
	AuNPs		Aptamer/CS	AFM1	Electrochemical signal	2–600 ng/L	0.9 ng/L	Milk, serum	[42]
	GNRs		Antibody	AFB1	Colorimetric signal	0.5–20 ng/ml	0.16 ng/mL	Peanut	[43]
Metal oxides and hydroxides	ITO	One-dimensional morphology, High electronic conductivity, Physicochemical stability, High specific surface area.	Antibody	AFB1	Electrochemical signal	0.125–1.5 ng/mL	0.15 ng/mL	-	[44]
	Sm_2_O_3_ nanorods		Antibody	AFB1	Electrochemical signal	10–700 pg/mL	57.82 pg mL^−1^ cm^−2^	-	[45]
	Fe_3_O_4_ nanoparticles		Antibody	AFB1	Electrochemical signal	0.05–5 ng/mL	0.07 ng/mL	Food	[46]
Carbon nanomaterials	CNTs	One-dimensional atomic sheet structure, Large surface area, Stable chemical properties, High electrical conductivity, Mechanical strength.	Antibody/ CdTe QDs	AFM1	Electrochemiluminescence signal	1.0–1.0 × 10^5^ pg/mL	0.3 pg/mL	Milk	[47]
	CNTs		Antibody/ PDDA	AFB1	Electrochemical signal	0.05–25 ng/mL	0.03 ng/mL	Rice	[48]
	c-MWCNTs		Antibody	AFB1	Electrochemical signal	0.25–1.375 ng/mL	0.08 ng/mL	-	[49]
	SWCNTs		Antibody/α-NP	AFB1	Electrochemical signal	0.01–100 ng/mL	3.5 pg/mL	Corn meal	[50]
	MWCNTs		Antibody/[BMIM]PF_6_	AFB1	Electrochemical signal	0.1–10 ng/mL	0.03 ng/mL	Olive oil	[51]
	MWCNT		Enzyme	AFB1	Electrochemical signal	3.2 × 10^−9^–721 × 10^−9^ moL/L	1.6 × 10^−9^ moL/L	-	[52]
	MWCNTs		MIP	AFB1	Electrochemical signal	1 × 10^−10^–l × 10^−5^ mol/L	0.03 nmol/L	Rapeseed oil, hogwash oil	[53]
	GO	Two-dimensional carbon nanomaterials, Excellent optical and electrical properties, High surface area, High strength and toughness, Good heat conduction performance, Facile functionalization and biocompatibility.	Antibody	AFB1	Electrochemical signal	0.5–5 ng/mL	0.23 ng/mL	-	[54]
	GO		Aptamer	AFM1	Electrochemiluminescence signal	5–150 ng/mL	0.01 ng/mL	Milk	[55]
	rGO		Antibody/PPy-PPa	AFB1	Electrochemical signal	10 fg/mL–10 pg/mL	10 fg/mL	-	[56]
	rGO		Aptamer/ polyaniline	AFB1	Electrochemical signal	1.0 × 10^−17^–1.0 × l0^−15^ g/mL	2 × 10^−18^ g/mL	Wine	[57]
	Graphene		Antibody/ PANI	AFB1	Electrochemical signal	0.05–25 ng/mL	0.034 ng/mL	Rice	[58]
	GQDs	Zero-dimensional atomic, Good dispersion, More abundant active sites, Good biocompatibility and photostability, Better chemical and physical properties, High water solubility.	Antibody/α-cyclodextrin	AFM1	Electrochemical signal	0.015–25 mmol/L	2 μmol/L	Milk	[59]
	CNHs		Antibody/L-F_3_O_4_-NFs	AFB1	Electrochemiluminescence signal	0.05–200 ng/mL	0.02 ng/mL	Corn	[60]
Other nanomaterials	MBs	High resistivity and permeability, Excellent sensitivity, Good dispersion and suspension, High binding rate, Controllable size, Easy functionalization.	Antibody	AFB1	Colorimetric signal	20–800 ng/L	12 ng/L	Corn	[61]
	MBs		Antibody	AFB1	Colorimetric signal	0.01–1 ng/mL	7 pg/mL	Maize	[62]
	MBs		Aptamer/DNA-scaffolded AgNCs	AFB1	Fluorescent signal	0.001–0.050 ng/mL	0.3 pg/mL	Wheat, rice, corn	[63]
	MBs		Aptamer/HCR	AFB1	Fluorescent signal	0.5–40 ng/mL	0.2 ng/mL	Peanut, milk	[64]
	MNPs		Aptamer/ PAPDI	AFB1	Fluorescent signal	-	0.01 nM	Maize	[65]
	QDs	Good light stability, Good biocompatibility, Long fluorescence life, Excellent sensitivity.	Aptamer/QDs	AFB1	Fluorescent signal	10–400 nmol/L	3.4 nmol/L	Rice, peanut	[66]

Note: NPs: nanoparticles; PEDOT: poly(2,3-dihydrothieno-1,4-dioxin); AFM1: aflatoxin M1; GNRs: gold nanorods; CS: complementary strand; ITO: indium tin oxide; PDDA: poly dimethyl diallyl ammonium chloride; c-MWCNTs: carboxylated multiwalled carbon nanotubes; SWCNTs: single-walled carbon nanotubes; [BMIM]PF_6_: 1-butyl-3-methylimidazolium hexanuorophosphate; MIP: molecular imprinted polymer; GO: graphene oxide; rGO: reduced graphene oxide; PPy: polypyrrole; PPa: pyrrolic acid; PANI: polyaniline; GQDs: graphene QDs; MBs: magnetic beads; MNPs: magnetic nanoparticles; HCR: hybridization chain reaction; PAPDI: perylene probe.

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
