# Peer review of "Recent Advances in Aflatoxins Detection Based on Nanomaterials"

_nanomaterials, 2020, doi:10.3390/nano10091626_

Round 1
Reviewer 1 Report
The review paper of Chunlei Yan et al. tries to give an overview on the recent advances in aflatoxins detection based on nanomaterials.
The topic is interesting, but the review has some lacks and needs major revision in order to be published.
Suggestions:
- Refer to the aflatoxins class in the plural form.
- Page 2 line 43-46. Is there any update since the 2002 monograph?
- Check the manuscript for errors like “NM” for nanometres, “PG” for pictograms, “Go” for graphene oxide, “Fe3O4”, etc.
- Table 1. Add other commonly used methods (for example ELISA method)
- Use a different form instead of “prenatal, post-natal” (page 2 line 48)
- In table 2, and in the manuscript in general, there are no references to lateral flow immunoassay based on nanomaterials for aflatoxins detection. This lack should be addressed. As you stated in the conclusion, most of the detection methods for AFs detection based on nanomaterials, reported in the literature, are still in the laboratory research stage. In contrast, several lateral flow immunoassay for AFs detection are already commercially available.
- Page 6 lines 175-178, please revise the sentence.
- Sections 2 and 3 have titles very similar (they mean exactly the same).
- I suggest to revise the review structure in order to organize the topic in a better and more logic way.
Reviewer 2 Report
This article describes the various AFB1 detection methods based on nanomaterials reported in the literature in the last decade. Some issues regarding the presentation of the results in the literature need to be addressed. Some important examples regarding nanomaterials used for AFB detection have not been discussed:
-In table 2, the column title is "Type of NPs" and the materials listed are not only nanoparticles but nanomaterials in general. In its current form the table is not informative and clear about the performance of each type of nanomaterial.
-In the column "detection signal", the detection technique used need to be specified instead of the generic name "Electrochemical signal", for example EIS (electrochemical impedance spectroscopy), DPV (differential pulse voltammetry). The linear range, sensitivity and the real samples tested should be specified in other columns. Also, many of these detection methods employ other components such as polymers or quantum dots and specific receptor molecules like antibodies or apatmers. These should also be mentioned, either as a separate column or together with the nanomaterial.
-Section 2.1. should also mention the increase of available binding sites for biomolecules on the surface of metal nanoparticles as an advantage.
-Section 2.2. should talk more about the physical properties and the advantages of using metal oxides and hydroxides.
Round 2
Reviewer 1 Report
The authors addressed most of the issues that emerged after the first review and improved their paper. However, a few new errors (resulting from the manuscript revised version) should be solved before publication.
“immune colloidal gold technique (GICT)”, I suggest to use one of the most commonly used term to refer to this technique, such as lateral flow immunoassay (LFIA) or also gold nanoparticles-based LFIA if you want to focus the attention to the use of nanomaterials.
Introduction, page 2 lines 82-85. As you did for HPLC, I suggest adding some more references for the other techniques (so that you have at least 2 references for each technique). For examples:
For ELISA, a very recent paper: Di Nardo, F.; Cavalera, S.; Baggiani, C.; Chiarello, M.; Pazzi, M.; Anfossi, L. Enzyme Immunoassay for Measuring Aflatoxin B1 in Legal Cannabis. Toxins 2020, 12, 265.
For gold nanoparticles-based LFIA (GICT in your revised paper), reference 18 and a recent paper: Di Nardo, F.; Alladio, E.; Baggiani, C.; Cavalera, S.; Giovannoli, C.; Spano, G.; Anfossi, L. Colour-encoded lateral flow immunoassay for the simultaneous detection of aflatoxin B1 and type-B fumonisins in a single Test line. Talanta 2019, 192, 288-294.
For fluorescence immunoassay, a very recent paper: Su, R.; Tang, X.; Feng, L.; Yao, G.-L.; Chen, J. Development of quantitative magnetic beads-based flow cytometry fluorescence immunoassay for aflatoxin B1. Microchemical Journal 2020, 155, 104715.
Check for some recent papers for chemiluminescence immunoassay and immunosensors.
Regarding the references, please use the same style for all of them.
Table 1. I appreciate that you added other techniques for aflatoxins detection, as suggested. However, the advantages and disadvantages of ELISA and LFIA need revision. For ELISAs, the detection speed is not fast, since they require several hours in order to obtain a result (while LFIA is fast). Moreover, they are not so cheap (while LFIAs are very low cost). An important advantage of ELISAs is that they allow analysing several samples at the same time. The repeatability of ELISAs is not so bad (while it is for LFIA), etc. I suggest revising the table accordingly.
Page 6 line 124. Please specify which aflatoxin (it is aflatoxin B1).
I am still not 100% sure regarding the review structure, since a lot of example reported in Section 2 are biosensor based on nanomaterials for aflatoxins detection (title of Section 3).
Pay attention to the use of the term “food detection”, its meaning is different from what you mean in the text.
Round 3
Reviewer 1 Report
The authors have addressed all the reviewer concerns, improving their manuscript.
This manuscript is a resubmission of an earlier submission. The following is a list of the peer review reports and author responses from that submission.